# Intention to comply with solid waste management practices among households in Butajira town, Southern Ethiopia using the theory of planned behavior

**Semu Debebe Fikadu[1], Abinet Arega Sadore[2], Gizachew Beykaso Agafari[2], Feleke Doyore Agide[2] \***

**1** Guragie Zone Health Department, Hossana, Ethiopia, **2** School of Public Health, College of Medicine and Health Sciences, Wachemo University, Hossana, Ethiopia

\* feledoag@yahoo.com

## Abstract

### Background

One of the world's most serious environmental issues is solid waste management. It is critical for researchers to understand the intention to comply with solid waste management. Thus, we aim to determine the intention to comply with solid waste management practice among households in Butajira Town using the Theory of Planned Behavior.

### Method

A community-based cross-sectional study was conducted on a sample of 422 households in Butajira from June 1 to June 30, 2020. The constructs and principles of the theory of planned behavior (TPB) were measured. We selected using a systematic sampling method and collected data by using a structured interviewer-administered questionnaire. Data was analyzed using SPSS version 25.0. The predictors of intention to practice solid waste management were identified using a multivariable linear regression model. A P-value of less than 5% was considered to declare a significant association.

### Results

The findings demonstrated that intention to practice solid waste management explained 86% of the variance explained by all predictors. The perceived behavioral control construct had the greatest impact on households' behavioral intentions to comply with solid waste management practice ($\beta$ = 0.16; CI (0.14, 0.18), followed by attitude ($\beta$ = 0.15; CI (0.11, 0.21) and subjective norms ($\beta$ = 0.12; CI (0.06, 0.17).

### Conclusion

Our study also found that intention has a substantial influence on the behavior of solid waste management practices. Therefore, there is a need to enhance service utilization for solid waste management and to improve outdoor solid waste dropping behavior through door-to-

**Data Availability Statement:** All relevant data are within the article and its Supporting Information files.

**Funding:** The authors received no specific funding for this work.

**Competing interests:** The authors have declared that no competing interests exist.

door collection services by municipality. Furthermore, further longitudinal research should be done through intervention mapping.

## Introduction

Solid waste management (SWM) in an urban area is a complex activity that involves the collection, transportation, recycling, resource recovery, and disposal of solid waste generated in an urban area [1]. Municipal solid waste is made up of various wastes generated by households and institutions such as schools, hospitals, slaughterhouses, and public restrooms [2]. Municipal waste is not well managed in developing countries due to the alarmingly increasing solid waste production, which is more than the capacity of the cities and municipalities. It was reported that waste collection rates are often lower than 70% in low-income countries, and more than 50% of the collected waste is often disposed of through uncontrolled land filling [3].

Globally, 2.6 billion people, or 39% of the world's population, do not have access to better sanitation [4, 5]. About 1.1 billion people continue to practice inappropriate solid waste management. In most developing countries, open, unregulated dumps are still the most common means of waste disposal. Many health-related problems result from improper waste management in urban areas [6, 7]. Environmental phenomena such as water, soil, and air pollution have been blamed on insufficient solid waste management [7]. It may also have a negative impact on one's health, the climate, and finances. The results of a report linking 22 diseases to poor solid waste management have been released by the US Public Health Service. Every year, 1.5 people die as a result of the inability to address these issues [8]. Municipal waste is poorly managed in developing countries due to alarmingly increasing solid waste production that exceeds the capacity of cities and municipalities. According to studies, waste collection rates in low-income countries are often below 70%, and more than half of the waste generated is disposed of by unregulated land filling [9, 10].

Since Ethiopia is one of the developing countries, the urban areas have a problem with solid waste management, which has its own negative impact on the environment. For instance, the study conducted on the assessment of Addis Ababa city revealed that improper and insufficient solid waste management is causing serious environmental problems and its management and associated environmental impacts are worthy of intention [10]. Ethiopia is one of the low-income countries facing the consequences of improper solid waste management. It was reported that about 20 to 30% of the waste generated in Addis Ababa, the capital city, remains uncollected [11]. However, rapid growth of the urban population and solid waste management are some of the main challenging problems for developing countries, and the waste disposal habits of the community cause the deterioration of the environment [6, 8]. Similarly, the study found that solid waste management activities at the household level in Wolaita Sodo town were lacking in terms of expressing the benefits of solid waste management practices, and that spatial coverage was hampered by a slew of serious issues [12].

Butajira is an old town with many public and private hospitals, health centers, industries, hotels, and small-scale enterprises where lots of solid waste is generated. The town municipality is mainly responsible for solid waste management in the town as there is no private organization involved in such tasks. There is no communal solid waste container deployed in different sites of the town. As a result, solid waste produced by every household is collected on the roadside. Even though solid waste management is supposed to be one of the critical public problems in Butajira, there was no study done.

Thus, it is critical to recognize the main determinants of behavior in order to modify human behavior since this can be strengthened or changed if the correct determinants that formulate particular behavior are identified [12, 13]. Tested theories are very important in explaining the healthy behaviors of individuals and communities and are used to develop evidence-based educational programs that can help individuals comply with certain behaviors [14, 15]. Thus, the current study uses the theory of planned behavior to assess the intention of solid waste management. The model was built with direct and indirect measurements. The direct measurements are attitude, normative beliefs, and perceived behavioral control. The model is apparent that it is open to the inclusion of other variables if they increase the predictive utility of the model [16]. However, as it has been criticized by many scholars in hypertension, blood donation and many more researches, the theory of planned behavior studies with results conflicting with TPB assumptions (e.g., null correlations between variables hypothesized to be highly related) rarely question the validity of the theory, but instead consider other explanations such as the operationalization of their study measures. Others have also questioned whether the hypotheses derived from the model are open to empirical falsification or whether they are essentially common-sense statements which cannot be falsified [16, 17]. In parallel speaking, since intention is the best and immediate predictor of behavior (solid waste management), normative beliefs and more perceived behavioral control improvements are still a day away from being established in Ethiopia and the study area in particular. That is why the TPB was applied to households to look into the factors influencing the intention of the community at household level waste management [14–17]. Therefore, this study aims to determine the intention and its predictors to comply with solid waste management practice among households in Butajira Town in Ethiopia using the Theory of Planned Behavior (TPB).

### Research hypothesis

There is a significant relationship between the TPB constructs (attitude, subjective norm, and Perceived Behavioral Control (PBC) and the households' intentions to comply with solid waste management practices.

## Materials and methods

### Study area and period

The study was conducted in Butajira town, in the Southern Nation's Nationalities and Peoples Region of Ethiopia, from June 1 to 30, 2020. It is 90 kilometers east of Wolkite town and 130 kilometers south of Ethiopia's capital, Addis Ababa. The rainy season lasts from June to October, with the rest of the year being mostly dry. The town has a total area of 9 km² and is administratively divided into 5 kebeles. According to the Butajira town health department's estimated population for 2019, the town had a total population of 52,962 and households of 10,808. Rural to urban migration, in addition to natural growth, led to the town's high population growth rate.

### Study design and populations

A community-based cross-sectional study preceded by elicitation study was conducted to assess intention to comply with solid waste management practice among households in Butajira Town in Ethiopia using the TPB. Since the study uses the theory TPB as a conceptual framework, elicitation study was conducted before proceeding to the main study in order to identify salient beliefs in the study population. According to TPB's recommendation, ten to twenty individuals from a given population were chosen to conduct an elicitation study; we

purposefully chose 15 households, of which 12 participated in the elicitation interviews. Participants were given a description of the behavior in terms of target, action, context, and time and were asked a series of open-ended questions to elicit beliefs and each in-depth interview lasted between 45–60 minutes.

## Elicitation study

It was done to elicit important psychographic beliefs regarding the intention to perform particular behavior (i.e. intention to comply with solid waste management practice in this case), the significant others and control beliefs for this intention. Purposive sampling was used to select the sample of 15 households for the elicitation study because it comes before the main study. The sample of 12 households was determined due to data saturation after the 10th respondent when a similar trend of answers started to appear, with no new data emerging and inferences regarding the topic being confirmed. This involves simultaneously collecting and analyzing data before deciding which data to collect next, who to interview next, and how many participants need to be included. In the present study, 12 participants (households) participated in the elicitation interviews.

## Sample size and sampling procedures

Then, the sample size for the main study was determined using a single population proportion formula with a confidence interval of 95% and a margin of error (d) of 5%. The estimate of predicted variance in intention to SWM practice was taken to be 50% since there was no evidence of the value of P on similar issue of the current study area) (p = 50%). Finally, to account for contingencies such as non-response or recording error, the sample size was increased by 10%, resulting in 384 X 10/100 + 384 = 422. The total sample size (422 people) was distributed proportionally based on the number of households in each kebele. Then, from each kebele, sample households were chosen using a systematic random sampling method. The interval value (K) was calculated for selected kebeles by dividing the total number of households of each selected kebele to the proportional sample size of the kebele. The initial household to be interviewed was selected randomly through a lottery method. If the selected housing units were changed to non-residential houses or just demolished for other purposes, the next housing which served as a residential house was selected.

## Measurement, variables and operation definitions

The outcome of this study is the intention to comply with solid waste management practices. Intention to solid waste management practice was measured by using three items (1. I intend to implement a sustainable "solid waste management practice in the households, for the next 12 months, 2. I am determined to implement a sustainable "solid waste management practice in the households, for the next 12 months" and I have decided to implement a sustainable "solid waste management practice in the households, for the next 12 months") on semantic differential scales (SDS). The composite score was done by summing up all the three items. The benefit/outcome of implementing solid waste management practice activity in the next 12 months was measured using four items on semantic differential scales (SDS) on bipolar adjectives (words with opposite meaning). Then, a composite score of direct attitude was obtained by summing up all the four items. Eight items were used to measure behavioral belief with responses ranging from unlikely at all (-3) to very likely (3). Evaluation of solid waste management practice belief was measured by asking participants to evaluate eight salient consequences accruing from implementing solid waste management practice. Each behavioral belief was multiplied by the outcome evaluation score to generate a new variable (indirect attitude)

representing the weighted score for each behavioral belief. A composite score of an indirect attitude was obtained by summing up all the eight products of behavioral belief and outcome evaluation. Three semantic differential scales (SDS) items were used to measure direct subjective solid waste management practice norm. A composite score of the direct subjective norm was obtained by summing up all the three items. To assess the indirect subjective norm toward solid waste management practice, participants were asked to complete five Likert scale items indicating how much they thought their family, neighbors, community volunteers, politicians/conservatives, and health professionals would appreciate their solid waste management practice implementation. Similarly, we weighted each normative belief by the score for motivation to comply with belief. Then; the composite scores of indirect subjective norm were created by summing up of the weighted beliefs. The direct measure of perceived behavioral control was measured by using six items on bipolar differential scales. A composite score of direct perceived behavioral control was obtained by summing up all the six items. Five control belief items were used to measure indirect perceived behavioral control ranging from strongly disagree to strongly agree scale and perceived power of control was measured using five items on the bipolar Likert scale ranging from very difficult to very easier scored on the −3 to + 3 scale response format. The control belief items were multiplied by those of perceived power of control of the beliefs. The composite score of indirect perceived behavioral control was calculated by summing these product scores together. Higher scores indicate a greater value for all measured T constructs toward the implementation of solid waste management practice in all cases. Then, each construct was operationalized as follows: Intention to solid waste management practice was operationally defined by items on a continuous measurement scale with a range of -9 to 9. The score was derived by calculating the sum total of questions 50, 51, and 52 from the questionnaire. The survey responses were coded as follows: 3 = extremely disagree; 2 = quite disagree; 1 = slightly disagree; 0 = neutral; 1 = slightly agree; 2 = quite agree; 3 = extremely agree. Thus, higher positive scores indicate a higher intention to practice solid waste management, while lower positive scores indicate a lower intention to practice solid waste management. Attitude toward solid waste management practice was operationally defined from the responses to items on a continuous measurement scale with a range of -12 to 12. Response choices to the survey questions were recorded using a 7-point scale and coded as -3 = extremely bad; -2 = quite bad; -1 = slightly bad; 0 = neutral; 1 = slightly good; 2 = quite good; 3 = extremely good. Higher positive scores would indicate a positive attitude toward the intention of engaging in the targeted solid waste management practice. Lower scores would indicate a less positive attitude. Items on a continuous measurement scale with scores ranging from -9 to 9 were used to operationally define the subjective norm. Response choices to the survey questions were coded as -3 = extremely disapprove; -2 = quite disapprove; -1 = slightly disapprove; 0 = neutral; 1 = slightly approve; 2 = quite approve; 3 = extremely approve. Higher positive scores would be associated with high social pressure/expectation in relation to the intention to engage in solid waste management practice. Lower scores would indicate little or no social pressure/expectation. Perceived behavioral control effects on solid waste management practice, as measured by items on a continuous measurement scale ranging from -18 to 18. The survey responses were coded as follows: 3 = extremely disagree; 2 = quite disagree; 1 = slightly disagree; 0 = neutral; 1 = slightly agree; 2 = quite agree; 3 = extremely agree. Higher positive scores would be associated with a strong belief in one's ability to perform or exert control over the behavior; influencing one's intention to engage in solid waste management practice. Normative control beliefs toward complying with the solid waste management practice and a subjective norm about compliance was operationally defined by items on a continuous measurement scale with a range of -45 to 45. Higher positive scores would be associated with high social pressure/expectation related to intention to comply with the solid waste

management practice and subjective norms about compliance. Waste segregation is the sorting and separation of waste types to facilitate recycling and correct onward disposal (dividing waste into dry and wet). Home compositing is the process of using household waste to make compost at home. In other words, composting is the biological decomposition of organic waste by recycling food and other organic materials. Waste collection is a part of the process of waste management in order to transfer solid waste from the point of use and disposal to the point of treatment or landfill (recycling, incineration).

## Data collection procedure and quality control

The questionnaire was initially developed in English by reviewing available literatures and guidelines [14–19] and translated to Amharic, and then back-translated to English by another person to maintain its consistency and modified based on the result of elicitation study. The training was given to data collectors and supervisors. Before the actual data collection, a questionnaire was pre-tested on 5% of the final sample size to test the clarity of the data collecting tools and to ensure respondent understanding of the questionnaire and the discrepancies was corrected and managed accordingly. Supervisors and investigators performed immediate supervision by the time of data collection to cross-check consistency, accuracy, and completeness of the collected data on daily basis. In addition to constructs of TPB, the questionnaire covered socio-demographic information.

## Data entry, processing and analysis

All collected data was entered into Epi-data version 3.1 and exported to SPSS version 25.0 for analysis. Descriptive analysis was used to see frequency distribution, mean and standard deviation. Correlation analysis was done between indirect and direct TPB variables to see the correlation between them. Multiple linear regression analysis was computed to test the strength and direction of association between the dependent variable and independent variables. First, bivariate regression analysis was employed. Then, multivariate regression analysis was applied following bivariate analysis with a p-value of $< 0.25$. An unstandardized β coefficient was used to interpret the effect of predictors on the intention of solid waste management practice. β is unstandardized coefficients which means original units besides the slope and tell if the independent variable is a significant predictor of the dependent variable. Beta is a standardized coefficient between -1 to +1 in range and show the strength of the prediction. Unlike standardized coefficients, which are normalized unit-less coefficients, an unstandardized coefficient has units and a 'real life' scale. An unstandardized coefficient represents the amount of change in a dependent variable intention to solid waste management due to a change of 1 unit of independent variable (socio-demographic variables and others). In other words; unstandardized coefficients are obtained after running a regression model on variables measured in their original scales. Standardized coefficients are obtained after running a regression model on standardized variables (i.e. rescaled variables that have a mean of 0 and a standard deviation of 1). $R^2$ was used for the ability of explanatory variables to explain dependent variables. Multicollinearity assumptions were tested by the variance inflation factor (VIF) and the value of all variables was below ten. Similarly, the tolerance value of all variables was above 0.10. Variables with a p-value of less than 0.05 at 95% confidence intervals were considered as statistically significant.

## Ethical considerations and informed consent

We obtained ethical clearance from Wachemo University School of Graduate Studies on 18 May 2020 (Ref. no: WCU/SGS/1173/12). On 19 June, 2020, a written permission from the Butajira town health office was obtained, and a support letter was sent to all selected kebele

(Ref. no: B/T/O/2234/2012). The right to self–determination and autonomy of all participants was respected. Thus, oral consent was obtained from all the respondents after explaining of the purpose of the study, risk/discomfort, benefits to the subject, and confidentiality of records, right to refuse participation and terminate participation in the study at any time. The informed verbal consent was obtained from the respondents after explaining the purpose of study. The study subjects had a right to withdraw at any time from the study. After data collection, a handbook on family health package was disseminated for each study participant by data collectors after orienting its purpose. The participants were assured of confidentiality with regard to all information acquired.

## Results

### Socio-demographic characteristics of participants

A total of 422 participants have participated with a response rate of 100%. Most of them (63.7%) were females and the participants' age ranged from 18 to 65 years. The mean age of participants was 31 years. Nearly 46% were orthodox Christian in religion and 58.80% were Gurhage in ethnicity (Table 1).

### Correlation between intention and constructs of theory of planned behavior

To explore the association between dependent and independent variables, all the necessary bivariate analysis was done using Pearson correlation. The Pearson's correlation coefficients showed that all the direct measures of the TPB except subjective norm were significantly and positively correlated with each other and with their respective indirect measures. From Table 2, the highest and lowest positive correlation was observed between intention and perceived behavioral control (r = 0.85, p < 0.001), and between intention and subjective norm (r = 0.29, p < 0.001), respectively. Most importantly, the indirect measures of attitude and perceived behavioral control were positively and significantly correlated with the corresponding direct measures. Finally, indirect measures of both attitude and perceived behavioral control were positively and significantly correlated with each other. The highest inter-indirect measure correlation was observed between indirect attitude and indirect perceived behavioral control (r = 0.38, p <0.001). The mean score of direct attitude, subjective norm and PBC were 7.94 (SD = 4.11), 5.48 (SD = 2.93) and 3.30 (SD = 12.63) respectively and intention with a mean score of 5.14 (SD = 4.21). There was a higher attitude score of 7.94 (SD = 4.11) towards solid waste management practice among households. For the PBC subscale, the participants' scores were at a moderate level (mean = 3.30 (SD = 12.63), with the relatively higher SD value suggesting limited consensus of opinion (Table 2).

### Magnitude and predictors associated with intention to solid waste management practice

Prior to the analysis, the assumptions of linear regression were checked. Then simple linear regression was performed to assess the association of each independent variable with the intention of solid waste management practice at a 95% confidence interval. Variables which were significant in simple linear regression were entered into multiple linear regressions for further statistical significance. Socio-demographic variables (age, sex, marital status and occupation) and all direct measures of theory of TPB variables were candidates for multiple linear regression models.

**Table 1. Socio-demographic characteristics of participants Butajira town, Southern Ethiopia (N = 422).**

| Variable | Categories | Frequency | Percent (%) |
|---|---|---|---|
| **Age** | 18–25 Years | 120 | 28.4 |
| | 26–35 Years | 151 | 35.8 |
| | 36–45 Years | 101 | 23.9 |
| | above 45 Years | 50 | 11.8 |
| | Total | 422 | 100.0 |
| **Gender** | Male | 153 | 36.3 |
| | Female | 269 | 63.7 |
| | Total | 422 | 100.0 |
| **Ethnicity** | Gurhage | 248 | 58.8 |
| | Amara | 74 | 17.5 |
| | Siltie | 36 | 8.5 |
| | Oromo | 45 | 10.7 |
| | Others | 19 | 4.5 |
| | Total | 422 | 100.0 |
| **The family size of households** | 1–3 Members | 116 | 27.5 |
| | 4–6 Members | 192 | 45.5 |
| | 7–8 Members | 77 | 18.2 |
| | Above 8 | 37 | 8.8 |
| | Total | 422 | 100.0 |
| **Religion** | Orthodox | 196 | 46.4 |
| | Muslim | 173 | 41.0 |
| | Protestant | 30 | 7.1 |
| | Catholic | 23 | 5.5 |
| | Total | 422 | 100.0 |
| Educational status | Unable to read and write | 100 | 23.7 |
| | Primary education (grade 1–8) | 165 | 39.1 |
| | Secondary education (grade 9–10) | 69 | 16.4 |
| | Preparatory education (grade 11–12) | 39 | 9.2 |
| | Diploma | 32 | 7.6 |
| | 1st degree | 12 | 2.8 |
| | 2nd degree | 5 | 1.2 |
| | Total | 422 | 100.0 |
| **Marital status** | Married | 266 | 63.0 |
| | Single | 117 | 27.7 |
| | Widowed | 22 | 5.2 |
| | Divorced | 17 | 4.1 |
| | Total | 422 | 100.0 |
| **Occupation of the participants** | Daily laborer | 79 | 18.7 |
| | Farmer | 7 | 1.7 |
| | Merchant | 217 | 51.4 |
| | Students | 46 | 10.9 |
| | Gov't employee | 37 | 8.8 |
| | Others | 36 | 8.5 |
| | Total | 422 | 100.0 |

Then, multiple linear regressions were performed, and variables from the first TPB, such as direct attitude, direct subjective norm, and direct perceived behavioral control, were included in the regression. These variables explain the model 76.0%. Then socio-demographic variables

**Table 2. Descriptive statistics and correlations of TPB constructs among households of Butajira town, Southern Ethiopia, 2020 (N = 422).**

| TPB Constructs | Behavioral Intention | Attitude (Direct) | Attitude (Indirect) | SN (Direct) | SN(Indirect) | PBC (Direct) | PBC (Indirect) | Scale Mean | Scale SD |
|---|---|---|---|---|---|---|---|---|---|
| Behavioral Intention | 1 | | | | | | | 5.14 | 4.21 |
| Attitude (Direct) | 0.65** | 1 | | | | | | 7.94 | 4.11 |
| Attitude (Indirect) | 0.67** | 0.71** | 1 | | | | | 28.37 | 13.66 |
| Subjective norm (Direct) | 0.29** | 0.11* | 0.16** | 1 | | | | 5.48 | 2.93 |
| Subjective norm (Indirect) | 0.30** | 0.26** | 0.18* | 0.09 | 1 | | | 13.08 | 9.36 |
| PBC (Direct) | 0.85** | 0.62** | 0.64** | 0.21** | 0.29** | 1 | | 3.30 | 12.63 |
| PBC (Indirect) | 0.55** | 0.32** | 0.38** | 0.14** | 0.16** | 0.40** | 1 | 9.56 | 12.25 |

Notes: **$p < .001$

*$p < .05$

(age, sex, marital status and occupation) of solid waste management practice were added to the theory constructs and explained the model 86%. The standardized regression coefficients, perceived behavioral control was found to be the best factor ($\beta = 0.16$; CI (0.14, 0.18) followed by attitude ($\beta = 0.15$; CI (0.11, 0.21). This means that a unit positive change in a household's perception of their ability to control circumstances that prevent them from implementing solid waste management practices increases the intention to implement solid waste management practices by 16% while other conditions remain constant. This finding was also supported by an elicitation study finding. One of the participants agreed that I think that support and encouragement would enable me to implement a solid waste management practice in the households, for the next 12 months. Similarly, a unit positive change in a household's attitude toward the benefits of implementing a solid waste management practice increases the individual's intention to implement it by 15% if all other factors remain constant.

This finding is supported by the findings of an elicitation study, in which one of the participants stated that solid waste management practice will help in getting a healthier life or that solid waste management practice contributes to preventing death due to poor waste management. This finding is likely to increase the intention of solid waste management practice. In contrast, a negative attitude expressed by one of the participants, such as I believe solid waste management practice is a potential source of disease, will eventually lead to a lower likelihood of solid waste management practice intention. Also, households who perceive significant others will approve of their implementing solid waste management practice will have a 12% higher intention to implement solid waste management practice than their counter parts. In this study, subjective norms was found to be the least significant factor associated with intention to solid waste management practice implement ($\beta = 0.12$; CI (0.06, 0.17). This finding was also supported by the findings of the elicitation study, with one of the participants agreeing, "I believe my neighbor wants me to engage in solid waste management practice, and I believe my family will be pleased to know I am practicing solid waste management activity." Such approval perception from a significant referent will help solid waste management practice. In the same way, standardized regression coefficients of socio-demographic variables (age above 46 years, occupation, farmer and students) were found to be a significant factor (Table 3).

Further analysis revealed that the indirect measures of the theory constructs probed the parameter estimates, behavioral beliefs→Attitude (Standardized Coefficient of Beta = 0.71; p < 0.05), show that aggregated behavioral beliefs have a significant influence in predicting attitude to comply with the solid waste management practice. Similarly, aggregated control

**Table 3. Multiple linear regression of intention to solid waste management practice and its predictors among households of Butajira town, Southern Ethiopia, 2020 (N = 422).**

| Variable | | Unstandardized B | Standardized β | t-value | Sig. | 95% CI for B |
|---|---|---|---|---|---|---|
| Constants | | 2.99 | | 7.37 | 0.00 | [2.19, 3.79] |
| Age | 26–35 years (ref.) | | | | | |
| | 18–25 years | -0.48 | -0.05 | -1.40 | 0.16 | [-1.15, 0.19] |
| | 36–45 years | -0.35 | -0.04 | -1.14 | 0.25 | [-0.94, 0.25] |
| | Above 46 years | 0.89 | 0.07 | 2.52 | 0.01 | [0.19, 1.59]* |
| Sex | Male (ref.) | | | | | |
| | Female | -0.38 | -0.04 | -1.66 | 0.10 | [-0.82, 0.07] |
| Marital status | Married (ref.) | | | | | |
| | Single | -3.12 | -0.33 | -10.36 | 0.00 | [-3.72, -2.53] |
| | Widowed | -0.59 | -0.03 | -1.12 | 0.26 | [-1.61, 0.44] |
| | Divorced | -0.03 | -0.01 | -0.07 | 0.94 | [-0.99, 0.92] |
| Occupation | Merchants(ref.) | | | | | |
| | Daily laborer | 0.27 | 0.03 | 0.87 | 0.39 | [-0.34, 0.87] |
| | Farmer | 2.22 | 0.07 | 3.43 | 0.00 | [0.95, 3.49]* |
| | Students | 1.07 | 0.08 | 3.14 | 0.00 | [0.40, 1.74]* |
| | Gov't employee | 0.60 | 0.04 | 1.79 | 0.07 | [-0.06, 1.26] |
| | Others | -0.48 | -0.03 | -1.34 | 0.18 | [-1.18, 0.23] |
| Direct attitude | | 0.15 | 0.15 | 6.27 | 0.00 | [0.11, 0.21]* |
| Direct SN | | 0.12 | 0.10 | 4.15 | 0.00 | [0.06, 0.17]* |
| Direct PBC | | 0.16 | 0.48 | 14.61 | 0.00 | [0.14, 0.18]* |

Notes: t-value > = 1.96 and P< 0.05 are Significant (*), ref. = reference

beliefs have a significant influence in predicting perceived behavioral control to comply with the solid waste management practice, control beliefs → perceived behavioral control (Standardized Coefficient of Beta = 0.40; p < 0.05). The aggregated normative beliefs show that normative beliefs → subjective norms (Standardized Coefficient of Beta = 0.09; p > 0.05) is not statistically significant in predicting subjective norm to comply with the solid waste management practice. Thus, it could be concluded that the belief constructs on the related domains could be used to explain the attitudes and perceived behavioral controls toward solid waste management practice for Butajira town households. Since the relationships were significant between the beliefs on the related attributes and attitudes, and perceived behavioral control, a direct pathway exists among those variables. Also, by considering the significant contributions that attitudes, subjective norms, and perceived behavioral control made on behavioral intentions, the following path model could be displayed in the present study.

$R^2$ values of 0.25, 0.50 and 0.70 can be interpreted as weak, moderate and substantial, respectively. Overall, the $R^2$ value of behavioral intention ($R^2$ = 0.86) indicates that the substantial amount of variance in solid waste management practice can be explained with perceived behavioral control, attitude and subjective norm. This finding suggests that the TPB can explain a significant amount of the behavioral intention to comply with solid waste management practice within the town environment. In general, the variance explained by the intention to solid waste management practice from all predictors was 86%. Findings show that attitude, subjective norm and perceived behavioral control have a direct positive effect on intention towards solid waste management practice. The indirect measures of the theory constructs probed the behavioral and control beliefs have a significant influence in predicting attitude and perceived behavioral control respectively, to comply with the solid waste

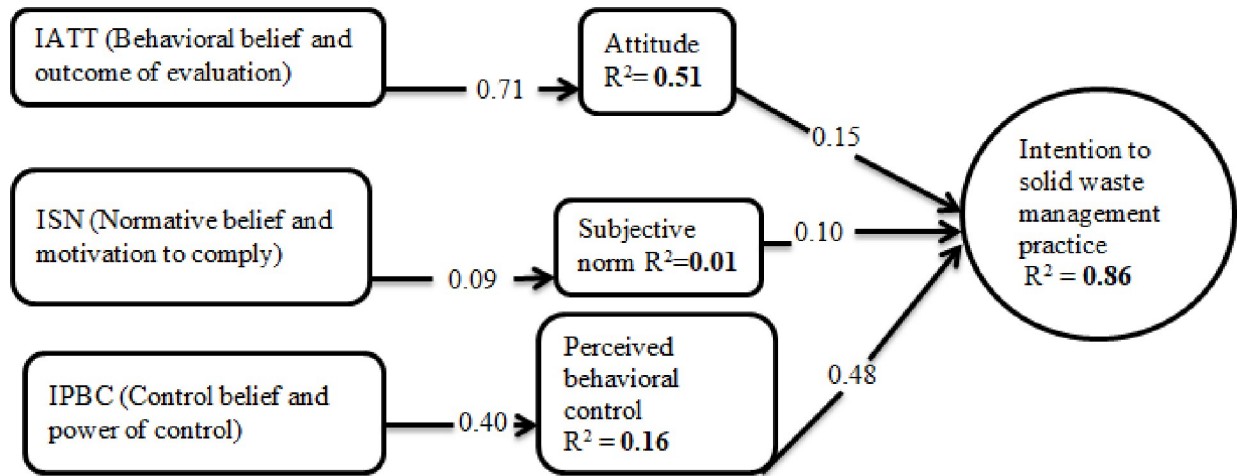

**Fig 1. Standardized path coefficient of theory of planned behavior variables (significant at p<0.05).** IATT = indirect Attitude; ISN = indirect Subjective norm; IPBC = indirect perceived behavioral control.

management practice. Fig 1 shows standard path coefficients' of theory of planned behavior variables which is significant at p<0.05: (i.e IATT = Indirect Attitude; ISN = Indirect Subjective Norms; IPBC = Indirect Perceived Behavioral Control). (Fig 1).

## Discussion

The model explained 86% of the households' intentions to comply with solid waste management practices. This finding is consistent with previous studies conducted in Ethiopia and abroad [13, 20], which found that the consequences of disposing of recyclable materials in separate waste bins have a significant direct relationship with intention. Correspondingly, this finding is higher than the study conducted in Bristol city (UK) and China [20–22]. This difference may be due to the study population, as those who visit health institutions may have a higher chance of getting access to information than their counter parts. The recent strong efforts for disease prevention and control programs being done by the Ethiopian government might have made a contribution to this difference. In contrast, the results from Bule Hora Town were not supported by the current study [16], which found that solid waste management practice consequences had a negative result on the household's intention to engage in waste management activities.

The result of our finding revealed that there is strong interplay between direct subjective norm and intention to solid waste management practice. Previous studies from Ethiopia, China and Cambodia also documented similar findings [15, 22, 23]. This suggests that decisions made in this context do not only concern the participants but also family, neighbors, health professionals and community volunteers. The current study found that eligible individuals are more likely to participate in this activity if their significant others are involved in solid waste management practice. This finding is also consistent with a previous study conducted in Taiwan [24], which found that subjective norms had a positive and significant influence on waste reduction intentions.

Previous research in China-case studies of Hangzhou, Ghana, and Canada discovered that the easier it is to recycle, the more likely it is to engage in a positive attitude [6, 21, 25], which is similar to the current study that revealed a strong interplay between perceived behavioral control and intention. This is also similar to the assumption TPB that suggests encouraging individuals to the aim of practicing a given behavior should involve consideration of factors

that are under their control. That is, those individuals who are able to control their fear of causing a potential source of disease and taking up too much time as a result of implementing solid waste management practices, which may lead to an increase in their intention to implement solid waste management practices. The current study also revealed that households do not agree on waste management practices that take up too much time and waste time. This result is consistent with previous study in Indonesia [7] which discovered that attitude, subjective norm, and perceived behavioral control all are influencing attitude. This shows that having knowledge of solid waste management practice can increase the perceived behavioral control towards intention. This is also consistent with Ghana's study [6] where the knowledge of knowing how to solid waste management practice significant towards is perceived behavioral control and correlated with intention.

This study found that providing households with recycling facilities and local collections holds great promise for improving households' intention to recycle. The finding is consistent with the study conducted in Nigeria [26]. Also, the study conducted in the Somalia Region in Ethiopia [13] shows that lack of resources and technical knowledge could be one of the factors that hinder waste management activities.

The results of the present study also showed that attitude is the strong predictor towards solid waste management practice among the households in Butajira town. This is consistent with studies conducted in Kenya, Nigeria, Malaysia and China [9, 22, 26, 27]. This might be due to those individuals who believed that solid waste management practice could lead a healthier life, prevent death related to poor waste management, prevent the risk of infection, promote the quality of the urban environment were more likely to have a favorable attitude and then more intended to solid waste management practice. This finding is also consistent with previous studies such as the study conducted in Karan District, Mogadishu, Somalia [28], which stated that environmental knowledge and attitudes can influence perception of households their awareness and susceptibility to the community-based project. This is similar to the assumption of TPB that states informing and raising awareness about a certain behavior (environmental issues), as well as proper solid waste management, are critical to the success of the goal of recycling household waste.

The present study revealed that external variables such as most socio-demographic variables had no significant effect on intention to solid waste management practice. This finding is consistent with a previous study conducted in Kenya [9], which stated that no significant relationship was found between household waste management practices and socio demographic characteristics. Also, the study conducted in Phnom Penh, Cambodia [23] showed that the influence of external factors was much smaller. Generally, the current study demonstrated that the TPB can explain a significant amount of behavioral intention compliance with solid waste management practice. This is similar to studies of various research scholars [20, 22].

As a limitation, firstly, the findings of the study reveal that the theory constructs significantly influences household waste management practice. This might not be the same as actual behavior. Secondly, the findings of the research are limited to urban communities (i.e. Butajira town). This may limit its applicability to rural areas like Southern Ethiopia. Thirdly, the present study did not include monetary incentive and moral obligation as predictors of household waste separation intention. Lastly, the cross-sectional design does not determine causality; this may result in a chicken-egg dilemma.

## Conclusion

The study revealed that intention has a substantial influence on the behavior of solid waste management practice. The study also found that the behavioral intention to solid waste

management practice was a function of the TPB psychographic variables; attitude, perceived social pressure and perceived behavioral control. Therefore, behavior change interventions should focus on increasing knowledge, perceived power that enable households to evaluate their control belief positively and empower them to develop ability against social norms that could compete with solid waste management practice and build an attitude that supports the behavior. Further study employing longitudinal design should be conducted to see the translation of behavioral intention to the actual behavior there by establishing a cause-effect relationship.

## Limitation of the study

Causal inferences can never be drawn out of the findings since the study is a cross-sectional one. During interview, there may be a social desirability bias.

## Supporting information

**S1 File.**
(DOCX)

**S2 File.**
(DOCX)

## Acknowledgments

We gratefully acknowledge the study participants for their active and volunteer participation. We are very thankful to urban health extension workers and kebele administrators for their support during data collection. We also gratefully acknowledge the contributions of the town health office and the municipality of the town for their unreserved information provision and the provision of one motorcycle for supervision of the data collection process for this research project.

## Author Contributions

**Conceptualization:** Semu Debebe Fikadu, Abinet Arega Sadore, Feleke Doyore Agide.

**Data curation:** Semu Debebe Fikadu, Abinet Arega Sadore, Gizachew Beykaso Agafari, Feleke Doyore Agide.

**Formal analysis:** Semu Debebe Fikadu, Abinet Arega Sadore, Feleke Doyore Agide.

**Funding acquisition:** Semu Debebe Fikadu.

**Investigation:** Semu Debebe Fikadu, Feleke Doyore Agide.

**Methodology:** Semu Debebe Fikadu, Abinet Arega Sadore, Gizachew Beykaso Agafari, Feleke Doyore Agide.

**Project administration:** Semu Debebe Fikadu, Gizachew Beykaso Agafari, Feleke Doyore Agide.

**Resources:** Semu Debebe Fikadu, Abinet Arega Sadore, Feleke Doyore Agide.

**Software:** Semu Debebe Fikadu, Feleke Doyore Agide.

**Supervision:** Semu Debebe Fikadu, Abinet Arega Sadore, Gizachew Beykaso Agafari, Feleke Doyore Agide.

**Validation:** Semu Debebe Fikadu, Abinet Arega Sadore, Gizachew Beykaso Agafari, Feleke Doyore Agide.

**Visualization:** Semu Debebe Fikadu, Abinet Arega Sadore, Gizachew Beykaso Agafari, Feleke Doyore Agide.

**Writing – original draft:** Semu Debebe Fikadu, Feleke Doyore Agide.

**Writing – review & editing:** Semu Debebe Fikadu, Abinet Arega Sadore, Gizachew Beykaso Agafari, Feleke Doyore Agide.

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
