## [Decision Letter · Decision Letter 0]

26 Nov 2021

PONE-D-21-28599Intention to Comply With Solid Waste Management Practices among Households in Butajira Town, Ethiopia Using the Theory of Planned BehaviorPLOS ONE

Dear Dr. Agide,

Thank you for submitting your manuscript to PLOS ONE. After careful consideration, we feel that it has merit but does not fully meet PLOS ONE’s publication criteria as it currently stands. Therefore, we invite you to submit a revised version of the manuscript that addresses the points raised during the review process.

ACADEMIC EDITOR: Based on the comments given by reviewers, there are two key issues that need to be addressed prior to the consideration of publication. First, on the theoretical soundness of this paper, authors are required to justify why the old TPB was used, and clarifications should be provided for some terminologies used. Another main issue is about the methodology of the paper. These include the sampling, as well as analyses used. At the current moment, there are unclearness issues in terms of variables definitions,  selection of results (p value of more than .05). 

We look forward to receiving your revised manuscript.

Kind regards,

Gabriel Ling, PhD

Academic Editor

PLOS ONE

Journal Requirements:

3. Please provide additional details regarding participant consent. In the Methods section, please ensure that you have specified (1) whether consent was informed and (2) what type you obtained (for instance, written or verbal). If your study included minors, state whether you obtained consent from parents or guardians. If the need for consent was waived by the ethics committee, please include this information.

NO

Reviewers' comments:

Reviewer's Responses to Questions

**Comments to the Author**

1. Is the manuscript technically sound, and do the data support the conclusions?

Reviewer #1: Partly

Reviewer #2: Partly

Reviewer #3: Yes

2. Has the statistical analysis been performed appropriately and rigorously? 

Reviewer #1: No

Reviewer #2: No

Reviewer #3: Yes

3. Have the authors made all data underlying the findings in their manuscript fully available?

Reviewer #1: No

Reviewer #2: Yes

Reviewer #3: Yes

4. Is the manuscript presented in an intelligible fashion and written in standard English?

Reviewer #1: No

Reviewer #2: Yes

Reviewer #3: Yes

5. Review Comments to the Author

Reviewer #1: Dear author,

Thank you for your submission. The manuscript focuses on an interesting topic. However, the main issues with this manuscript are; (1) What is the main contribution of this study to the body of knowledge?, contextual contribution is not acceptable. (2) Why the old theory of planned behavior (TPB) was used instead of the new TPB? (3) Methodology part is not clear. Details of the comments as per attachment.

Reviewer #2: It is an interesting work in identifying the underlying factors which can influence the society behavior towards a better waste management practices. The methodology could be a bit unclear, which can require some definitions on the variables involved and some elaborations on the regression analysis. Some suggestions:

(1) Kindly include line number and section numbers for easier referencing, if applicable.

(2) Introduction: the literature review can be further enriched to highlight previous works and their observations with similar or different approaches, which can then better highlight the research gap and objective of the current work.

(3) Introduction: would be good to define what is included under the solid waste management activities/practices to be complied with? e.g. waste segregation, recycling or home composting? Is there any specific regulation/policy on waste management practices by the country/city?

(4) Materials and Methods: measurement definitions: what are the 3 items and 4 items etc referring too? suggest can elaborate briefly or in brackets for further clarity. This section is quite lengthy. Kindly consider breaking into shorter paragraphs and with a flowchart diagram showing the progression of scoring, if applicable.

(5) Materials and Methods: variables definitions: suggest that the variables can be further elaborated and defined. e.g. what does it mean or is included under indirect attitude?

(6) Materials and Methods: data entry and analysis: why does a p-value of less than 0.25 was chosen? As co-linearity among parameters could pose some issues for regression analysis, is there any approaches/strategies in addressing this?

(7) Results : pg 10: the r= 0.29 or 0.38 seems not to be a high correlation strength... Is there some mechanisms/literature to discuss further on the relationship?

(8) Results: pg 16: some of the observations were reported by only 1 participants. Would be good also if the discussion can be added on with more representative observations.

(9) Discussions: if any sensitivity analysis considered or to be considered that might affect the observations ?

(10) table 3 pg 18: where does 0.71 derived from as it was not observed in table 3? why does the variable with beta 0.09 was chosen as it was stated as not significant in the text? Would be good to discuss/clarify the selection of variables for the final analysis.

Reviewer #3: The article takes a systematic random sampling method to select a research sample with size of 442, and ensure that the statistics to obey normal distribution. The results of data analysis are consistent with the conclusions of research. That is, attitude, subjective norms, and perceived behavior control have a significant effect on the intention of solid waste management practice. In the descriptive statistical analysis of the sample, the manuscript describes the distribution of socio-demographic characteristics of participants, and also analyzes the association between research variables by Pearson correlation. Finally, this article uses linear regression model to analyze the effects of direct and indirect attitude, subjective norms and perceived behavior control on the intent of implementing solid waste management practices. Overall, the study further expanded the field of research in the field of solid waste and focus on a micro-subject (family or individual). But there are some problems as following.

In the section of Introduction, the authors elaborated on the importance of studying the practical problems of household solid waste management and the rationality of choosing the theory of planned behavior as the theoretical basis for the study. However, it is also important to illustrate the current status of municipal solid waste management in Ethiopia through statistical data, which can help readers understand why Ethiopian cities can be used as research objects. In addition, it is recommended to elaborate on the reasons for choosing Butajira Town as the specific research area in the Study area and period section.

In the section of Study design and populations, the authors use elicitation study to identify salient beliefs in the research population. This is reasonable, but the selection process of the research objects of the elicitation study should be explained.

In the section of Ethical considerations and informed consent, the authors obtained the permission from the Butajira health office on June 7, 2012, but the research period is from June 1 to 30, 2020. Please confirm that this time of permission is correct. Too long a time interval will cause confusion, and it is recommended to delete the content of permission.

Attitude, subjective norms, and perceived behavior control are the three important elements of the theory of planned behavior. For this study, not only must clearly define these three elements but also the meaning of intention variables must be clearly explained. In the section of Measurement, variables and operation definitions, the authors use three items on the Semantic Difference Scale to measure the intention of the solid waste management practice, but fails to explain the content of the items in the scale, which leads to understanding the meaning of variables difficultly. Similarly, although the authors divided attitude, subjective norm, and perceived behavior control into two categories of direct and indirect, they did not present the content of the relevant items in the scale on the manuscript. In order to understand the meaning of the variables easily, it is recommended to select appropriate items in the scale to explain the variable.

In the section of Result, the authors use frequency and percentage to show the socio-demographic characteristics of the sample due to the existence of categorical variables. This processing method is reasonable. However, if there are relevant statistical data, the socio-demographic characteristics between the sample and the population should be compared to further prove the representativeness of the sample.

In the descriptive statistics and correlation analysis in Table 2, all the direct measures of the TPB except subjective norm were significantly and positively correlated with each other and with their respective indirect measures. It is necessary to explain why there is no correlation between the direct and indirect measures of subjective norms.

In the latter part of Result section, the authors use multiple linear regression to test the influence of direct attitudes, direct subjective norms and direct perceptual behavior control on the family's intention to implement solid waste management practice. However, in the subsequent further analysis, the results shown in Figure 1 need to analyze the direct impact of IATT, ISN, and IPBC on the household's intention to implement solid waste management practice. For example, in addition to affecting Attitude and then Intention, IATT may have a direct and significant positive impact on Intention.

In the Discussion section, the authors mentioned that cross-sectional design cannot determine causality. To solve this problem, in addition to designing a longitudinal study, it is also possible to build a causal diagram based on previous research literature to identify the causal relationship between variables.

Finally, the authors should state their research hypothesis in the appropriate place of the manuscript.

6. PLOS authors have the option to publish the peer review history of their article (what does this mean?). If published, this will include your full peer review and any attached files.

Reviewer #1: No

Reviewer #2: No

Reviewer #3: No

---

## [Author Response · Author response to Decision Letter 0]

3 Jan 2022

Date 27 December, 2020

Dear Academic Editor and Reviewers, 

Thank you so much for your valuable comments and interest in the publication of the manuscript. Your comments have improved the quality of our manuscript and almost all the comments are incorporated into the current manuscript. The "Revised Manuscript with Track Changes" will show how much we improved our manuscript as per your comment.

We are pleased to have an opportunity to revise our manuscript, entitled "Intention to Comply with Solid Waste Management Practices among Households in Butajira Town, Southern Ethiopia Using the Theory of Planned Behavior." [PONE-D-21-28599]. In the revised manuscript, we have carefully considered the editors' and reviewers’ comments and suggestions. As instructed, we have attempted to succinctly explain the changes made in reaction to all comments. We reply to each comment in point-by-point fashion. We have color-coded the revised manuscript as text. The responses to the concerns raised by the editor and reviewer are below and are italic coded as follows. The editor’s and reviewers’ comments were very helpful overall, and we are appreciative of such constructive feedback on our original submission. After addressing the issues raised, we feel the quality of the paper has greatly improved.

Note: Datasets used and analyzed during the current study are available from the corresponding author on reasonable request.

Responses to Editor

Q##: Please ensure that your manuscript meets PLOS ONE's style requirements, 

Response: Thank you for the comment. We corrected the manuscript in order to meet PLOS ONE's style requirements.

Q##: Please include additional information regarding the survey or questionnaire

Response: We thank you so much for your comments and refining our manuscript for publication. Both English and Amharic version questionnaire is added.

Q##: Please provide additional details regarding participant consent.

Response: Consent form is there in the manuscript and added as supplementary materials too.

Q##: Please clarify the sources of funding (financial or material support) for your study.

Response: Elaborated in the text of manuscript where appropriate.

Responses to Reviewer #1

Thank you for your valuable comments and interest on our manuscript for publication. Your comments were incorporated into the revised manuscript and clarified one by one.

Q##: What is the main contribution of this study to the body of knowledge? contextual contribution is not acceptable. 

Response: Thank you for the valuable comment. The outcome of this study will provide to the Federal Ministry of Health and other health organizations with a comprehensive understanding and strategies on the households’ behavioural intentions to be engaged in the solid waste management practice. The guiding principles extracted from the findings will be useful for researchers, policy-makers, practitioners and other public health experts in Solid Waste Management system.

Responses to reviewer #2:

Thank you very much for your comments and refining our manuscript for publication. Your comments were incorporated into the revised manuscript and clarified one by one.

Q##: Methodology could be a bit unclear. Introduction: would be good to define what is included under the solid waste management activities/practices to be complied with? e.g. waste segregation, recycling or home composting? Is there any specific regulation/policy on waste management practices by the country/city?

Response: Corrected. Thank you for the valuable comment.

Q##: Materials and Methods: measurement definitions: what are the 3 items and 4 items etc referring too? Suggest can elaborate briefly or in brackets for further clarity. This section is quite lengthy. Kindly consider breaking into shorter paragraphs and with a flowchart diagram showing the progression of scoring, if applicable.

Response: We thank you so much for the valuable comment. Since the constructs of theory of planned behaviour are scale variables, it is mandatory to use some items or parameters that can measure the constructs of theory of planned behaviour. So, these items refer some parameters that were used to measure the constructs.

Q##: Materials and Methods: variables definitions: suggest that the variables can be further elaborated and defined. e.g. what does it mean or is included under indirect attitude?

Response: Thanks for the comment. Attitude measurement is an attempt to convert observations of a person's behaviour toward a referent into an index representing the presence, strength, and direction of the attitude presumed to underlie the behaviour. Direct methods to measure attitude are those in which the respondents are either informed that their attitudes are being measured or are made aware of it by the nature of the attitude measurement technique. It is assumed that the respondent and the researcher view the experimental task similarly and have attached the same meaning and significance to the response that is requested. In contrast to the direct methods, indirect methods to measure attitude yield responses that are not taken literally. Rather, the respondent's performance on an overt and seemingly straightforward objective task is thought to unconsciously reveal latent psychosocial constructs that are interpreted as attitude.

Q##: Materials and Methods: data entry and analysis: why does a p-value of less than 0.25 was chosen? As co-linearity among parameters could pose some issues for regression analysis, is there any approaches/strategies in addressing this?

Response: We incorporated your valuable comment in the current manuscript. Variance Inflation Factors (VIFs) measures the extent to which multi-collinearity has increased the variance of an estimated coefficient. It looks at the extent to which an explanatory variable can be explained by all the other explanatory variables in the equation. The VIF for each independent variable can be obtained by regressing it against all others in the set being analysed, and then calculating (1/[1 − R2]). A VIF of 1.8 tells us that the variance of that predictor variable (i.e. its standard error) is 80% greater than would be the case with no collinearity effect: VIFs of 2.5 or greater are generally considered indicative of considerable collinearity suggesting that there will be difficulty in separating out the independent contribution of variables with such large VIFs—although some authors. Some strategies that can solve collinearity problems are: dropping a Redundant Variable, Transforming the Multicollinear Variables and Increasing the Sample Sizes. We used a p-value of less than 0.25 in bivariate regression analysis because most of the references used this cut off point to identify variables that are eligible for multiple linear regression analysis.

Q##: Results: pg 10: the r= 0.29 or 0.38 seems not to be a high correlation strength... Is there some mechanisms/literature to discuss further on the relationship?

Response: Well, thanks! To improve the correlation coefficient, one of the mechanisms is increasing the difference between the variables. This is done by identifying the independent variable observation, which is same or close to dependent observation value, and replacing it with the value which would increase the difference between the variables.

Q##: Results: pg 16: some of the observations were reported by only 1 participant. Would be good also if the discussion can be added on with more representative observations.

Response: Corrected with great thanks.

Q##: Table 3 pg 18: where does 0.71 derived from as it was not observed in table 3? Why the variable with beta 0.09 does was chosen as it was stated as not significant in the text? Would be good to discuss/clarify the selection of variables for the final analysis.

Response: Corrected with great thanks 

Responses to reviewer #3

Authors would like to thank you for your comments and refining our manuscript for publication. Your comments were incorporated into the revised manuscript and answered one by one.

Q##: It is also important to illustrate the current status of municipal solid waste management in Ethiopia through statistical data, which can help readers understand why Ethiopian cities can be used as research objects. In addition, it is recommended to elaborate on the reasons for choosing Butajira Town as the specific research area in the Study area and period section.

Response: Corrected 

Q##: the selection process of the research objects of the elicitation study should be explained.

Response: Corrected

Q##: In the section of Ethical considerations and informed consent, the authors obtained the permission from the Butajira health office on June 7, 2012, but the research period is from June 1 to 30, 2020. Please confirm that this time of permission is correct. Too long a time interval will cause confusion, and it is recommended to delete the content of permission.

Response: Corrected, June 7, 2012 was in Ethiopian calendar, changed to G.C. (June 19, 2020)

Q##: also the meaning of intention variables must be clearly explained

Response: We thank you for the comment and elaborated the meaning of intention variables.

Q##: In the section of Measurement, variables and operation definitions, the authors use three items on the Semantic Difference Scale to measure the intention of the solid waste management practice, but fails to explain the content of the items in the scale.

Response: Corrected

Q##: Although the authors divided attitude, subjective norm, and perceived behaviour control into two categories of direct and indirect, they did not present the content of the relevant items in the scale on the manuscript. In order to understand the meaning of the variables easily, it is recommended to select appropriate items in the scale to explain the variable.

Response: Corrected (we narrated this issue under measurement of variables in the manuscript). Thanks for the comment.

Q##: It is necessary to explain why there is no correlation between the direct and indirect measures of subjective norms.

Response: Corrected and explained in the text.

Q##: The authors should state their research hypothesis in the appropriate place of the manuscript.

Response: Stated with thanks!

 Thank you very much to editor and reviewers for your precious comments!

---

## [Decision Letter · Decision Letter 1]

24 Feb 2022

PONE-D-21-28599R1Intention to Comply With Solid Waste Management Practices among Households in Butajira Town, Southern Ethiopia Using the Theory of Planned BehaviorPLOS ONE

Dear Dr. Agide,

Thank you for submitting your manuscript to PLOS ONE. After careful consideration, we feel that it has merit but does not fully meet PLOS ONE’s publication criteria as it currently stands. Therefore, we invite you to submit a revised version of the manuscript that addresses the points raised during the review process.

ACADEMIC EDITOR: Kindly respond to reviewer 2's concerns before it can be considered for publication.

We look forward to receiving your revised manuscript.

Kind regards,

Gabriel Hoh Teck Ling, PhD

Academic Editor

PLOS ONE

Journal Requirements:

Reviewers' comments:

Reviewer's Responses to Questions

**Comments to the Author**

1. If the authors have adequately addressed your comments raised in a previous round of review and you feel that this manuscript is now acceptable for publication, you may indicate that here to bypass the “Comments to the Author” section, enter your conflict of interest statement in the “Confidential to Editor” section, and submit your "Accept" recommendation.

Reviewer #2: (No Response)

2. Is the manuscript technically sound, and do the data support the conclusions?

Reviewer #2: Yes

3. Has the statistical analysis been performed appropriately and rigorously? 

Reviewer #2: Yes

4. Have the authors made all data underlying the findings in their manuscript fully available?

Reviewer #2: Yes

5. Is the manuscript presented in an intelligible fashion and written in standard English?

Reviewer #2: Yes

6. Review Comments to the Author

Reviewer #2: The paper investigates the possible relationship between several aspects of TPB with solid waste management through statistical analysis. Thank you for addressing the comments. Some suggestions:

(1) Introduction: would suggest to elaborate briefly on the studies applying TNB on waste management, e.g. the approaches, findings or limitations on the respective studies, instead of having lumped references, so that the research gap and the objective of the current work can be better highlighted.

(2) Research hypothesis: abbreviation PCB not defined.

(3) Methodology- Study design & Measurement, variables and operation definitions: for the 'solid waste management practice' and 'sustainable solid waste management practice' mentioned, are there any specific activities being referring to? e.g. waste segregation, home composting, collection etc

(4) Methodology- Data entry, processing and analysis: how is unstandardized β coefficient calculated and why is it chosen over standardized β coefficient?

For correlation analysis, would suggest that the input for indirect and direct TPB variable can be stated for easier referencing. r is not mentioned here.

(5) under Ethical considerations and informed consent, the reference number for the support letter appeared to have three empty boxes. Is it a typo? Kindly check.

(6) R2 not subscripted. Kindly check.

(7) suggest the conclusion paragraph can be under a new section Conclusion.

7. PLOS authors have the option to publish the peer review history of their article (what does this mean?). If published, this will include your full peer review and any attached files.

Reviewer #2: No

---

## [Author Response · Author response to Decision Letter 1]

9 Apr 2022

Date: 08 April, 2022

Dear Academic Editor and Reviewers, 

We thank you so much for your valuable comments and interest in the publication. We revised the comments of reviewer 2. The "Revised Manuscript with Track Changes" will show how much we improved our manuscript as per your comment.

We are also pleased to have an opportunity to revise our manuscript, entitled "Intention to Comply with Solid Waste Management Practices among Households in Butajira Town, Southern Ethiopia Using the Theory of Planned Behaviour" [PONE-D-21-28599]. The comments are very important to improve the quality of paper. We used track-changes to reply each concern as per your suggestion, and put below one by one. We believe the paper's quality has considerably improved after we addressed all of the issues identified.

Responses to Editor

• We considered the points you suggested to meet PLOS ONE's style requirements while you are preparing the manuscript.

N.B: Datasets used and analyzed during the current study are available from the corresponding author on reasonable request.

Responses to Reviewer #2

Thank you for your valuable comments and interest in our manuscript for publication. Your comments were incorporated into the revised manuscript and clarified one by one.

Q (1). Introduction: would suggest elaborating briefly on the studies applying TNB on waste management, e.g. the approaches, findings or limitations on the respective studies, instead of having lumped references, so that the research gap and the objective of the current work can be better highlighted. 

Answer: Your comment is incorporated in the revised document. Thank you for the valuable comment.

Q(2). Research hypothesis: abbreviation PCB not defined. 

Answer: We defined the abbreviation for the first time as Perceived Behavioural Control (PBC). Thank you for the comment.

Q(3). Methodology- Study design & Measurement, variables and operation definitions: for the 'solid waste management practice' and 'sustainable solid waste management practice' mentioned, are there any specific activities being referring to? e.g. waste segregation, home composting, collection etc.

Answer: Thank you for the concern and comments. Our research aim is to assess the intention of the households on the variables you listed under solid waste management, not to measure each variable in a practical manner. However, we put the operational definitions in the current document. 

Q(4). Methodology- Data entry, processing and analysis: how is unstandardized β coefficient calculated and why is it chosen over standardized β coefficient?

Answer: Thank you for the comment. It is really interesting. β is unstandardized coefficients which means original units besides the slope and tell if the independent variable is a significant predictor of the dependent variable. Beta is a standardised coefficient between -1 to +1 in range and show the strength of the prediction. Unlike standardized coefficients, which are normalized unit-less coefficients, an unstandardized coefficient has units and a 'real life' scale. An unstandardized coefficient represents the amount of change in a dependent variable intention to solid waste management due to a change of 1 unit of independent variable (socio-demographic variables and others). In other words; unstandardized coefficients are obtained after running a regression model on variables measured in their original scales. Standardized coefficients are obtained after running a regression model on standardized variables (i.e. rescaled variables that have a mean of 0 and a standard deviation of 1).

Q(5). Under Ethical considerations and informed consent, the reference number for the support letter appeared to have three empty boxes. Is it a typo? Kindly check.

Answer: We corrected in the revised manuscript. Thank you!

Q(6). R2 not subscripted. Kindly check.

Answer: Thanks for the comment. We corrected in the current document.

Q(7). Suggest the conclusion paragraph can be under a new section Conclusion.

Answer: Thank you so much for the comment. We put the conclusion paragraph in a new section.

 Thank you very much to editor and reviewers for your valuable comments!

---

## [Decision Letter · Decision Letter 2]

5 May 2022

Intention to Comply With Solid Waste Management Practices among Households in Butajira Town, Southern Ethiopia Using the Theory of Planned Behavior

PONE-D-21-28599R2

Dear Dr. Agide,

We’re pleased to inform you that your manuscript has been judged scientifically suitable for publication and will be formally accepted for publication once it meets all outstanding technical requirements.

Kind regards,

Gabriel Hoh Teck Ling, PhD

Academic Editor

PLOS ONE

Additional Editor Comments (optional):

Reviewers' comments:

Reviewer's Responses to Questions

**Comments to the Author**

1. If the authors have adequately addressed your comments raised in a previous round of review and you feel that this manuscript is now acceptable for publication, you may indicate that here to bypass the “Comments to the Author” section, enter your conflict of interest statement in the “Confidential to Editor” section, and submit your "Accept" recommendation.

Reviewer #2: All comments have been addressed

2. Is the manuscript technically sound, and do the data support the conclusions?

Reviewer #2: Yes

3. Has the statistical analysis been performed appropriately and rigorously? 

Reviewer #2: Yes

4. Have the authors made all data underlying the findings in their manuscript fully available?

Reviewer #2: Yes

5. Is the manuscript presented in an intelligible fashion and written in standard English?

Reviewer #2: Yes

6. Review Comments to the Author

Reviewer #2: The study investigates the intention to comply with solid waste management practice among households based on of Planned Behavior and the relationship among behavior predictors. The comments have been addressed.

7. PLOS authors have the option to publish the peer review history of their article (what does this mean?). If published, this will include your full peer review and any attached files.

Reviewer #2: No

---

## [Editor Report · Acceptance letter]

30 Jun 2022

PONE-D-21-28599R2 

Intention to Comply With Solid Waste Management Practices among Households in Butajira Town, Southern Ethiopia Using the Theory of Planned Behavior 

Dear Dr. Agide:

I'm pleased to inform you that your manuscript has been deemed suitable for publication in PLOS ONE. Congratulations! Your manuscript is now with our production department. 

Kind regards, 

on behalf of

Dr. Gabriel Hoh Teck Ling 

Academic Editor

PLOS ONE